# Structural Investigation of Therapeutic Antibodies Using Hydroxyl Radical Protein Footprinting Methods

**DOI:** 10.3390/antib11040071

**Published:** 2022-11-14

**Authors:** Corie Y. Ralston, Joshua S. Sharp

**Affiliations:** 1Molecular Foundry Division, Lawrence Berkeley National Laboratory, 1 Cyclotron Road, Berkeley, CA 94720, USA; 2Department of BioMolecular Sciences, University of Mississippi, Oxford, MS 38677, USA

**Keywords:** hydroxyl radical footprinting, structural mass spectrometry, fast photochemical oxidation of proteins (FPOP), flash oxidation (FOX), X-ray footprinting with mass spectrometry (XFMS), therapeutic antibody structure

## Abstract

Commercial monoclonal antibodies are growing and important components of modern therapies against a multitude of human diseases. Well-known high-resolution structural methods such as protein crystallography are often used to characterize antibody structures and to determine paratope and/or epitope binding regions in order to refine antibody design. However, many standard structural techniques require specialized sample preparation that may perturb antibody structure or require high concentrations or other conditions that are far from the conditions conducive to the accurate determination of antigen binding or kinetics. We describe here in this minireview the relatively new method of hydroxyl radical protein footprinting, a solution-state method that can provide structural and kinetic information on antibodies or antibody–antigen interactions useful for therapeutic antibody design. We provide a brief history of hydroxyl radical footprinting, examples of current implementations, and recent advances in throughput and accessibility.

## 1. Introduction

The design and commercial production of therapeutic antibodies has grown substantially in the last two decades as methods continue to advance in the areas of bioconjugates [1,2], affinity maturation and humanization of antibodies [3], glycoengineering [4], and phage display and other protein evolution methods [5]. In addition, next-generation sequencing and rapid DNA synthesis methods have helped to accelerate the timeframe of monoclonal antibody (mAb) production [6]. In 2021 alone, the commercial development of therapeutic antibody treatments to treat various diseases grew by 30% [7], even excluding those developed against the coronavirus SARS-CoV2, the development of which is proceeding at an accelerated rate. As of 2022, antibodies as therapeutics are currently the fastest growing class of biopharmaceuticals [8].

In parallel, high-resolution structural biology characterization methods continue to improve, with recent significant advances in cryoEM structural resolution [9] and continual advances in X-ray and electron diffraction methods [10], as well as nuclear magnetic resonance (NMR) methods [11], and thousands of structures of antibodies or antibody fragments are currently available in Protein Data Bank [12]. Other imaging methods, such as fluorescence microscopy, have continued to make improvements in achievable resolution [13], while computational tools are also widely used in antibody development to predict protein interfaces and assess biophysical characteristics such as immunogenicity or solubility. Protein modeling recently made a significant leap in successful structure prediction with AlphaFold [14] as well as RoseTTAFold [15]. While structural information is not strictly necessary for mAb discovery or maturation, it can significantly affect the success of mAb design and production in two areas. First, high-resolution structural information on the antigen epitope region, the antibody paratope region, or the interface between the two can accelerate the rational design of antibodies. For example, one of the earliest mAbs developed, Rituximab [16], was in use for years before the protein crystal structure of the antibody was solved in complex with an antigen fragment [17], providing at last the rationale for understanding Rituximab antigen binding and setting the stage for designing biosimilars based on the antibody. Second, mAb structure must be monitored during scale-up to ensure batch-to-batch consistency, to monitor solubility, and as a function of time to ensure product shelf life and continued efficacy.

However, the currently most accessible high-resolution protein structure determination methods, macromolecular crystallography (MX), NMR, and cryoEM, require specific sample preparations that prevent the use of these methods to monitor structure and dynamics in solution for mAb formulations. MX requires the crystallization of proteins; cryoEM is measured in the frozen state; NMR typically requires high concentrations and cannot be applied for protein sizes greater than 100 kDa [18]. In contrast, mass spectrometry-based methods do not pose these same issues, as they can be used for a range of protein sizes and concentrations and can be used to measure the dynamical properties of molecules in solution. For example, hydrogen deuterium exchange mass spectrometry (HDX-MS) is used to monitor protein backbone hydrogen bonding and is often used to map epitope binding regions on mAbs [19]. Similarly, footprinting mass spectrometry is used to measure side-chain accessibility at up to residue-level resolution, making it an ideal method to map binding regions between mAbs and their cognate antigens. Footprinting methods using the hydroxyl radical (˙OH) in particular have advanced substantially in the last decade in throughput and accessibility, with several commercial footprinting options now available [20,21].

Figure 1 shows conceptually how HRPF can be used for characterizing various aspects of antibody structure and binding. For example, recent work by Misra et al. used hydroxyl radical protein footprinting (HRPF) to investigate the effect of various buffers and excipients on the higher-order structure, thermal stability, and aggregation properties of adalimumab. Not only could conformational changes and aggregation be detected, but the specific regions of the protein affected could be determined [22]. Sharp et al. showed the effective determination of the adalimumab epitope in trimeric TNFα using the commercially available flash oxidation (FOX) protein footprinting system [20]. Schick et al. used HRPF to probe the primary epitopes for polyclonal anti-drug antibodies raised against a bi-specific mAb in a monkey pharmacological model [23], demonstrating the utility of hydroxyl radical footprinting not only in drug–target interactions but also in host drug response. Schoof et al. used HRPF in conjunction with crystallography and cryo-EM to delineate an ultrapotent nanobody interaction with the SARS-CoV2 spike protein [24], while Sevillano et al. used the method to improve the affinity maturation of a recombinant Fab inhibitor [25]. In examples of HRPF applied to glycosylated proteins, Deperalta et al. used the method to investigate a dimer interaction in a therapeutic monoclonal antibody [26], and Li et al. used HRPF to characterize the interaction between the glycosylated HIV gp120 protein with a broadly neutralizing antibody [27]. In two further examples of antibody interactions, Li et al. used the method to map the epitope of an antibody–interleukin-23 interaction [28], and Zhang et al. used HRPF to map the binding interface between a monoclonal antibody Fab fragment and the vascular endothelial growth factor protein [29]. HRPF was also used to help to characterize an anti-CCL-1 antibody drug candidate for acute myeloid leukemia [30]. For a more general review of HRPF usage in drug discovery, we refer the reader to Kiselar and Chance [31].

With this mini-review, we cover a brief history of hydroxyl radical footprinting methods, recent technical advances in the field, and practical considerations for using these methods, particularly for investigation of antibody–antigen interactions. This overview is not meant to provide a comprehensive review of footprinting methods for mAb development, but rather to highlight the application of HRPF to antibody structure and dynamics and provide the reader with a starting point to use these methods for therapeutic antibody characterization.

## 2. A Brief History of Hydroxyl Radical Footprinting Methods

The term “footprint” in the context of solvent accessibility was in use as early as the 1970s, when several studies highlighted the use of chemical or enzymatic digests to map out the protective “footprint” of a protein on DNA [32,33]. Not long after, studies pointed to using Fe-EDTA-generated hydroxyl radicals rather than small molecules or enzymes to provide higher-resolution footprinting results because ˙OH cleaves DNA at every base position with almost no sequence specificity [34,35]. Subsequently, different methods of producing ˙OH radicals for the purpose of nucleic acid footprinting were explored, including gamma rays [36] and X-rays [37]. Protein structural analysis using amino acid chemical modification [38] predates nucleic acid footprinting, though the term “protein footprinting” may not have come into widespread use until the 1980s [39,40]. In contrast to DNA or RNA footprinting *cleavage* reactions, protein hydroxyl radical footprinting relies on the *covalent modification* to protein side chains or backbone. As with nucleic acid footprinting, the hydroxyl radical is an excellent probe for the structural characterization of proteins, since its small size enables single-residue resolution, and many methods for generating hydroxyl radical in solution have been explored, including Fe-EDTA chemistry [41], X-rays [42], electron beam [43], and continuous-wave UV photolysis of H_2_O_2_ and UV laser photolysis of H_2_O_2_ in the method termed fast photochemical oxidation of proteins (FPOP) [44,45]. After exposure to ˙OH, proteins are digested, and peptide or residue modification rates are monitored using bottom-up liquid chromatography mass spectrometry (LCMS) methods [46]. These post-exposure analysis methods have also improved steadily through the years, allowing more complex and larger proteins to be investigated under a variety of conditions.

The use of ˙OH footprinting for protein structure, folding, and dynamic characterization has grown steadily through the years and is currently accessible through commercial FPOP and plasma-based systems [20,21,47], through General User proposals at synchrotron X-ray sources [48,49] (X-ray footprinting with mass spectrometry (XFMS)), and through bench-based Fenton chemistry [41]. These methods, which we collectively refer to as hydroxyl radical protein footprinting (HRPF), all have advantages and drawbacks in terms of sample preparation, timescales, access, and cost. In the next section, we discuss practical considerations for the use of these various methods.

## 3. Practical Applications and Considerations

In general, as proteins fold and/or interact with binding partners, the local solvent accessibility of regions within and on the surface of the protein change, and the apparent rate of hydroxyl radical modification of residues changes accordingly. All HRPF methods are based on this central principle, and all require LCMS methods to measure the extent of the modification of residues. Modification can be determined per residue or per peptide, depending on the level of resolution required for a given experiment. A general conceptual overview of HRPF is shown in Figure 2. The fraction of unmodified molecules, per residue or per peptide, is plotted against increasing exposure time, resulting in what are generally termed “dose response” plots. In general, the fraction modified is in the order of several percent, except sometimes, e.g., in the case of highly reactive residues such as methionine. As the solvent accessibility of the residue increases, the fraction of molecules containing that modified residue increases proportionally. As depicted in this example, the dose–response plots reveal which residues or peptides become more accessible or less accessible to solvent during events such as ligand binding, protein–protein interactions, or protein conformational changes. These changes, in turn, are used to infer structural information. In the hypothetical example shown in Figure 2, residues 1 and 2 become more protected (less solvent accessible), while residues 3 and 4 become more accessible during a protein conformational change, leading to the conclusion that one domain has rotated relatively to another.

In comparison to crystallography, NMR, and cryoEM, in which atomic coordinates are obtained, the data obtained using HRPF are in the form of solvent accessibility maps, generally on a per-residue basis. In practice, this information must then be mapped onto the protein sequence in order to infer structural information, as shown conceptually in Figure 2. Often, several experiments are performed, with both experimental and technical repeats, and a standard error analysis is performed on the dose–response plots to show the overall quality of the data. The collection of HRPF data can be performed quickly, often requiring only several hours of instrument time. However, the optimization of buffers can take several experiments, and the acquisition of LCMS data is dependent on access to an instrument. The data analysis portion of the study may take minutes to weeks, depending on the complexity of the protein system under study and the level of spatial resolution desired. A limitation of the method is that many standard buffers, such as Tris or HEPES, scavenge hydroxyl radicals with high efficiency. When such buffers are used, the background radical scavenging must be compensated for [50,51,52]. In practice, HRPF experiments are most commonly carried out with phosphate buffer or Na-cacodylate buffer, though some progress has been made with high-flux-density beamlines or higher-yield photolysis systems for overcoming scavenging effects. One advantage of the method is that with modern implementations of sample delivery, it becomes possible to couple sample delivery with additional characterization, such as spectroscopy [53], or with sample pre-processing, such as inline size-exclusion chromatography [54].

Each of the HRPF methods described below also shares similarities with respect to ˙OH reactivity with specific residues and with buffer constituents, and Table 1 summarizes high-level differences between the methods.

The extent of the modification of protein side chain and backbone is driven by both solvent accessibility and intrinsic reactivity to ˙OH. In practice, when comparing a protein in two different states (e.g., with and without ligand), the intrinsic reactivity effectively normalizes out between states and can be ignored. However, intrinsic reactivity does affect the residues most commonly detected using HRPF. Side chains are more reactive in general than the protein backbone [55,56], and among the side chains, the most reactive side chains are methionine, cysteine, tryptophan, tyrosine, and phenylalanine, while the least reactive are alanine and glycine [57]. The rate constants for the reaction of amino acids in solution with hydroxyl radicals have been measured relatively to each other in the order Cys > Trp > Tyr > Met > Phe > His > Arg > Ile > Leu, with others being less reactive [57], although it should be noted that residues in proteins necessarily have local sequence environments that affect the reactivity to hydroxyl radicals [58]. Methods for the normalization of protein reactivity have also been proposed, in which modification rates for residues are multiplied by a calculated intrinsic reactivity “protection factor” derived from free amino acids [59]. This intrinsic reactivity estimate has been found to be reliable for the most reactive amino acids, with effects of neighboring amino acids on intrinsic reactivity [58] becoming more significant for less reactive amino acids [60].

Furthermore, ˙OH reactivity to the buffer must be taken into account, since buffer constituents compete with protein for ˙OH, which is short-lived and highly reactive to molecules containing sulfur, aromatic groups, or significant numbers of C-H bonds. Imidazole or high concentrations of TRIS buffer, for instance, are very challenging to use in HRFP experiments, as they are highly scavenging of hydroxyl radicals. Sodium phosphate buffers are most commonly used when possible, as sodium phosphate is largely unreactive to ˙OH. When organic buffers must be used, minimizing buffer concentration and eliminating aromatics and unoxidized sulfurs is recommended.

The various HRPF approaches differ in the method by which hydroxyl radical molecules are produced, which in turn dictates experimental parameters, such as sample concentrations and buffers, the folding/interaction timescale, and cost.

### 3.1. Chemical ˙OH Footprinting

A lab-based approach is one of the most accessible hydroxyl radical footprinting methods, as it can be performed with standard reagents. Chemical means exist for producing hydroxyl radicals, including the use of peroxynitrous acid [61] and metal/H_2_O_2_ chemistry [62,63,64]. The latter can be accomplished, for example, with inexpensive lab reagents such as ferrous ammonium sulfate, EDTA, and H_2_O_2_ in the following reaction, known as Fenton chemistry [65]:Fe^2+^ + H_2_O_2_ → Fe^3+^ + OH^−^ + ˙OH(1)

Generally, when applying this reaction scheme, EDTA is used to chelate iron in order to increase the solubility of the metal, enabling reactions at neutral pH, and to prevent the direct binding of the metal to biomolecules in solution. Ascorbate is also used to reduce Fe^3+^ back to Fe^2+^, allowing hydroxyl radicals to be continuously produced. Chelated Fe-EDTA was first used by Tullius and coworkers to footprint DNA [66,67] and has been widely used since that time to footprint protein and/or nucleic acids, with recent methodology development of a method using high-density well plates [68]. While the use of Fenton chemistry for HRPF is straightforward, one disadvantage of this method is that it requires the incubation of protein with H_2_O_2_ for seconds or minutes, making the dynamics of biomolecules difficult to study. The slow timescale of Fenton chemistry can cause the probing of oxidation-induced artifactual conformations, necessitating strictly limited oxidation or alternative methods to ensure that native conformation is maintained [69,70]. In addition, the interaction of proteins with millimolar concentrations of H_2_O_2_ in solution can disturb protein structure [43] and/or cause modifications independent of hydroxyl radical-induced oxidations [71].

### 3.2. X-Ray-Based ˙OH Footprinting

Hydroxyl radical footprinting using synchrotron X-rays was first demonstrated in the late 1990s at National Synchrotron Light Source [37]. With this method, protein samples at low concentrations (typically micromolar) in buffer are placed in an X-ray beam for short irradiation times, typically milliseconds or shorter. Since the water concentration (55 M) is orders of magnitude higher than the protein concentration, the primary interaction of the X-rays is with the water in the solution and not directly with the protein molecules. Water is radiolysed to hydroxyl radicals on a timescale faster than microseconds [57]. The hydroxyl radicals in turn react with nearby protein side chains, and to some extent, protein backbone. Because hydroxyl radicals also quickly react with each other and other radical species, the central tenet of the experiment is that a protein side chain is only modified if a hydroxyl radical is produced from a water molecule in close proximity to the side chain; i.e., due to their extremely short lifetime, hydroxyl radicals do not diffuse throughout the solution during or after irradiation [72]. The higher the beam flux density is, the shorter the X-ray exposure time that is necessary to generate sufficient measurable modification to reveal the “water map” snapshot at the time of irradiation is. A typical X-ray-based HRPF experiment is shown in Figure 2.

The sample can be delivered via a capillary or jet nozzle, as is currently used at the Advanced Light Source (ALS) synchrotron footprinting beamline [48], or via a 96-well plate, such as is currently used at the National Synchrotron Light Source (NSLS-II) footprinting beamline [73] (Figure 3). In the capillary, jet, or 96-well plate setup, the velocity of the sample past the X-ray beam is used to control the duration of X-ray exposure, and for these experimental configurations, exposure times can be varied from microseconds to milliseconds, depending on the experiment parameters. For stationary tubes, a shutter must be used. For most synchrotron broadband X-ray beams that are used in the XFMS experiment, the shutter material must withstand substantial energy from the beam, up to 10^18^ photons/sec; therefore, typical fast shutters used in monochromatic beam applications are not possible and limit the reliable minimum opening time to the order of milliseconds. This shortcoming led to the development of flow delivery systems, as described in Section 4.

A recent extension of the X-ray footprinting experiment is the incorporation of trifluoromethyl radical labeling [74,75]. In this experiment, sodium triflinate is added to the solution prior to irradiation. During X-ray exposure, ˙OH radicals displace ˙CF_3_ radicals from sodium triflinate. The ˙CF_3_ radicals, in turn, modify the protein side chain with reactivity comparable to that of ˙OH, producing -CF_3_ adducts detectable by LCMS, but with different reactivity for various side chains. Notably, amino acids that are not reactive to hydroxyl radicals, such as alanine and glycine, can be modified with ˙CF_3_; a more detailed analysis of the relative reactivity of amino acids to hydroxyl or ˙CF_3_ is described in [75]. Thus, ˙CF_3_ labeling is highly complementary to the more standard ˙OH labeling and can be used simultaneously to enhance the sensitivity of the experiment and extend the resolution of the method.

### 3.3. Laser-Photolysis-Based ˙OH Footprinting

In the FPOP method, ˙OH radicals are generated via the UV laser photolysis of H_2_O_2_. With this lab-based method, protein samples are typically prepared in millimolar concentrations, injected via a syringe pump through a sample chamber containing H_2_O_2_, where the timed laser pulse generates ˙OH radicals for oxidative labeling, and then ejected into tubes containing a radical scavenger (Figure 4). Experiments can incorporate a mixing T in line with the sample exposure cell so that H_2_O_2_ can be rapidly injected into the cell immediately prior to laser exposure, thus limiting protein exposure to H_2_O_2_ to under a second [76]. For a protein of given size and buffer requirements, the scavenger, H_2_O_2_, and protein concentration should be optimized. In practice, the concentration of H_2_O_2_ must be optimized for OH modification and is typically in the millimolar concentration range [77]. An exclusion volume between laser shots allows one to limit most of the sample volume to irradiation by a single laser pulse, and typical hydroxyl radical exposure times in the FPOP experiment for a given volume of sample are in the order of microseconds. In practice, a radical scavenger is also added to the flow system in laser-based FPOP, where it controls the lifetime of the ˙OH radicals. This prevention of long-lived exposure to hydroxyl radicals allows protein to be thoroughly labeled without probing oxidatively unfolded conformations [70].

The FPOP method has also been used to investigate protein interactions in whole live cells [78] or even intact *C. elegans* nematodes [79]. In the case of cells, H_2_O_2_ was added to cell preparations and readily crossed the cell membrane. Many oxidative modifications require the presence of dissolved molecular oxygen [57], which makes X-ray and chemical footprinting on whole cells challenging because of the generally low intercellular oxygen content. However, the endogenous enzyme catalase breaks down H_2_O_2_ to produce O_2_, making FPOP a viable option for in vivo footprinting. For in vivo footprinting in *C. elegans*, the use of chemical penetration enhancers improves the uptake of hydrogen peroxide, allowing proteins to be more thoroughly labeled [80].

## 4. Recent Advances

Advances in automation and inline dose monitoring in the last several years have made HRPF methods more accessible and capable of higher throughput. Below, we describe several such advances, highlighting developments in the FPOP and XFMS fields in particular, though advances have recently also been made in plasma HRPF [21] and high-throughput Fenton chemistry [68].

### 4.1. Automation

#### 4.1.1. FOX System

A recent development is the FOX Photolysis System introduced by GenNext Technologies, Inc., as the first commercial benchtop HRF system (Figure 4B [20]). The FOX system uses a proprietary high-pressure flash lamp and a proprietary drive unit to generate a very brief (<10 µs FWHM) and very intense broadband UV output, which is used to drive the flash photolysis of hydrogen peroxide. While this method is very similar conceptually and in practice to FPOP, it is technically a different method that is referred to as flash oxidation (FOX). The FOX Photolysis System is a semi-automated sample oxidation system that consists of four modules: a fluidics module that drives sample introduction by flow injection; a photolysis module that contains the flash lamp and drive system used to photolyze hydrogen peroxide and form hydroxyl radicals; a dosimetry module that performs real-time inline radical dosimetry using a UV absorbance dosimeter (generally adenine, which is a nucleoside that can be used as a dosimeter [52,81] or Tris [51]); and a sample collector that collects only oxidized samples and diverts carrier buffer to waste.

For analyses on the FOX system, samples are pre-mixed with hydrogen peroxide (generally at 10–100 mM) and a radical dosimeter. The FOX system oxidizes proteins at lower peak ˙OH concentrations for longer times (several microseconds for FOX as opposed to ~1 µs for FPOP), eliminating the need for radical scavengers in solution. The sample is manually injected into a sample loop, and the sample loop is then switched in line with the flowing carrier buffer to drive the sample through the vertical photolysis path. The sample is pushed into an optical cell that focuses the light from the flash lamp, which operates at up to 2 Hz. Similar to FPOP, the sample flow rate and the rate of flash lamp repetition are timed to prevent any volume of sample from being illuminated by more than one flash, with a user-defined exclusion volume separating illuminated volumes to correct for laminal flow effects and diffusion. Illuminated samples next flow through the dosimetry cell, which measures the UV absorbance at 265 nm in real time. Samples can be compensated by adjusting the drive voltage of the photolysis unit in control software to increase or decrease the effective radical dose until the pre-determined radical dosimeter response is reached, compensating for any differential influences on the effective radical dose (e.g., differential radical scavenging, different effective hydrogen peroxide concentrations, different lamp strengths). Finally, a miniature fraction collector is used to only collect the sample that is exposed to the desired effective radical dose (as determined using radical dosimetry), with the remaining sample and/or carrier buffer being diverted to a waste collector. The sample is collected in tubes that are pre-loaded with a quench solution to eliminate hydrogen peroxide and secondary oxidants (most commonly catalase and methionine amide), leaving a protein sample that has been stably modified. Samples are sent for LCMS analyses, and an accompanying data processing package, FoxWare^TM^, is available from GenNext Technologies, Inc., for the automated analysis of LCMS data. The system can handle a wide variety of sample volumes, with analysis time per sample depending on the total sample volume analyzed. For 12 µL samples, samples take less than eight minutes each to label and quench, with all steps being automated except manual injection. This system was used to perform the epitope mapping of adalimumab, successfully mapping the epitope on trimeric TNFα [20] (Figure 4C).

#### 4.1.2. HTP 96-Well Plate and Software

The X-ray footprinting beamline at NSLS-II [49] recently implemented a 96-well PCR plate, in which the plate can be scanned across an X-ray beam, enabling a high-throughput data collection mode [73]. In this implementation, X-ray exposure is controlled via an upstream shutter and defined through a user GUI interface. The plate accommodates 8- and 12-tube PCR strips for ease of pre- and post- sample processing or fluorescence analyses. A custom recirculating water bath and Peltier cooling blocks allow experiments to be conducted from −40 to +37 °C. Because the sample environment is entirely contained within tubes, the system additionally allows the investigation of biosafety level 2 materials, viruses, or prion-like proteins to be conducted.

#### 4.1.3. Jet Delivery System and Software

A recent advance for the X-ray HRPF experiment is the development of a Rayleigh liquid-jet sample delivery system [48,82]. With this system, samples are loaded into a high-pressure syringe pump for driving samples through capillary tubing to a polished jet nozzle. The liquid sample is ejected from the nozzle in a continuous stream past the X-ray beam and collected in Eppendorf tubes containing quench solution in a fraction collector (Figure 3). The velocity of the sample determines the X-ray exposure time, and the jetting regime is defined by the flow velocity, exposure time required, and the beam-defining apertures, which also serve as scatter guards. Image analyses using a strobe light show a consistent jet profile up to 8 mm from the ejection point. Depending on the beam size and nozzle dimensions, exposures can range from tens to hundreds of microseconds. We note that slower speeds that would be needed for longer exposure times are not possible with the jet system because the jet would become unstable. However, with the jet “containerless” delivery, higher doses are achievable, since there is no glass or other material containing the sample; therefore, shorter X-ray exposure times are generally used. For instance, a calculation of absorbed energy at the ALS 3.2.1 bend magnet beamline showed that a jet with a diameter of 20 µm received a 10-fold increase in integrated photons relative to a standard capillary delivery system [48]. Nozzles producing jet streams of different diameters are possible, and to date, 20, 50, 75, 100, and 200 µm jets have been used and characterized [82]. Sample exposures, volume collected, and other experimental parameters are controlled via a LabView-based GUI, facilitating the ease of the experiment. The jet delivery system provides an alternative to the more standard capillary delivery method and has some advantages, namely, shorter exposure times using very-high-brightness beams become possible with the jet, since there is no burning of capillary material. More recently, the jet/capillary sample endstation was implemented at beamline 3.3.1 at ALS and is available for use as part of ALS and Molecular Foundry General User Programs at Lawrence Berkeley National Laboratory.

### 4.2. Inline Dosimetry

#### 4.2.1. UV Absorbance Dosimeters (FPOP/FOX)

Hydroxyl radical dosimetry is the measurement of the effective radical dose, that is, the amount of hydroxyl radical the analyte is exposed to, including both the amount of hydroxyl radical generated and the amount of hydroxyl radical consumed by the non-analyte components of the sample. An ideal radical dosimeter has a quantitative response to a broad effective radical dose and is easily measurable in real time to monitor HRPF reactions as they occur. Reviews of dosimetry methods using reporter peptides or small molecules are covered elsewhere [81,83]. For the high hydroxyl radical fluxes generated using FPOP, the first reported radical dosimeter with an optical readout was adenine [81].

Adenine and its associated nucleosides/nucleotides strongly absorb UV light with a local maximum of ~265 nm. This absorbance is not affected by the presence of hydrogen peroxide during the FPOP timescale; however, the reaction of adenine with hydroxyl radicals generates products that have greatly decreased UV absorption at 265 nm [81]. This response to the effective radical dose is linear across a wide range of experimental conditions [81]. As the experimental readout is a single-wavelength UV absorbance measurement, the effective radical dose can be measured in real time with a relatively simple capillary UV absorbance spectrophotometer [52]. This real-time radical dosimetry makes possible the correction not only for the effects of variations in hydrogen peroxide concentrations or light-source fluence on hydroxyl radical generation, but also for the effects of different sample additives or buffers on hydroxyl radical scavenging, ensuring that the changes in the observed HRPF footprint are due to changes in the higher-order structure (Figure 5) [52]. For example, in the analysis of mAbs and other biopharmaceuticals, this real-time dosimetry allows the scavenging effects of different drug formulations to be compensated, enabling the study of the effects of formulation on mAb aggregation and higher-order structure to be conducted [22]. Real-time dosimetry also allows the scavenging effect of mAb in epitope-mapping experiments to be compensated, ensuring that the epitope sees equivalent doses of hydroxyl radical in the presence or absence of mAb [20].

While adenine is the most popular real-time dosimeter for FPOP and FOX HRF, any molecule that shows a significant change in optical properties upon reaction with hydroxyl radicals can potentially be used as a real-time radical dosimeter, so long as the change in optical properties is quantitative and linear (or at least, predictable) across the range of the experimental conditions used. An example of an alternative radical dosimeter that has been applied successfully is tris(hydroxymethyl)aminomethane, a standard biological buffer more commonly called Tris. Unlike adenine, Tris is normally UV transparent but gains significant UV absorbance at ~265 nm upon reaction with hydroxyl radicals, probably via the conversion of the alcohol to an aldehyde and/or imine [51]. It is probable that numerous other small molecules exist that could act as hydroxyl radical dosimeters, giving researchers options for the dosimeter best suited for their system.

#### 4.2.2. Inline Fluorescence Monitoring in XFMS Experiments

As with any ˙OH experiment, in the XFMS method, it is important to optimize the X-ray dose for the particular buffer system in use, since common buffers and additives scavenge ˙OH to different extents. A recent advance in the rapid determination of the dosage is the incorporation of inline fluorescence excitation in the XFMS experiment (Figure 6) [82]. In this implementation, a diode light source impinges on the sample as it traverses into the fraction collector, and the resulting fluorescence is collected in real time via a fiber-optic and photomultiplier tube (PMT). The excitation light can be positioned immediately after the X-ray impingement point, enabling the collection of fluorescence spectra from incorporated dye molecules, such as Alexa-488, which fluoresces in the visible light range. This dye is commonly added to buffers prior to an XFMS experiment to empirically assess the degree of scavenging by a given buffer system, since Alexa-488 fluorescence decreases with the increase in exposure to hydroxyl radicals [84]. The use of visible light excitation/emission is also preferable to UV, since UV-absorbance-based dosimeters suffer from background signals and artifacts. While Alexa-based dosimetry is useful in XFMS experiments, it does not work well in the FPOP environment, giving complex non-linear responses [81]. In previous implementations of the XFMS experiment, tubes containing Alexa-488 in buffer were exposed to an X-ray beam, and fluorescence was measured for each tube using a stand-alone fluorimeter outside the experimental hutch. In the new implementation, Alexa-488 fluorescence is converted to a voltage measurement via the PMT and plotted in real-time via a LabView-based software control interface, enabling the rapid determination of the scavenging environment of a given buffer system to be conducted, so that X-ray exposure times can be adjusted accordingly.

The new implementation additionally opens up the possibility of measuring intrinsic fluorescence from proteins when the diode/PMT path is positioned immediately “upstream” of the X-ray impingement point. In this case, fluorescence can provide a global readout of the protein structure or state of folding, which is highly complementary to the residue-specific structural information obtained with HRPF.

## 5. Outlook

All oxidative footprinting methods comprise two essential components: (1) the covalent modification of proteins where side chains are accessible and (2) the quantification of the oxidations through LCMS. This brief review summarizes the methods used to produce oxidative modifications, focusing on the methods of FPOP and XFMS in particular, both of which have made substantial advances in recent years in throughput, automation, ease of use, and reliability. The stability of HRPF labeling allows HRPF to be combined with powerful protein chemistry techniques (such as tryptic digestion, enzymatic deglycosylation, and sample clean-up methods) and allows modified samples to be archived and re-analyzed after proteolysis. HRPF labeling is now available as a commercial system and at national user facilities. While these advances in instrumentation represent a significant advance for the field, the HRPF method has not yet achieved the same widespread use in the industry as HDX, for example, in part because the second component of the method—LCMS—is not yet seamlessly integrated into the system, as it is for commercial HDX systems. For XFMS experiments at DOE-supported national synchrotrons, staff often conduct LCMS post-irradiation if users do not have access to their own mass spectrometry instrumentation. In other cases, users send samples for LCMS analyses to mass spectrometry centers on a cost basis.

Nonetheless, HRPF is growing in recognition and has been used to obtain important information related to antigen-binding sites, epitope mapping, host antibody responses, and antibody aggregation in a number of antibody systems. The field will likely continue to expand as the methods for oxidative labeling become capable of yielding even higher throughput and as LCMS methods and instrumentation become more accessible.

## Figures and Tables

**Figure 1 antibodies-11-00071-f001:**
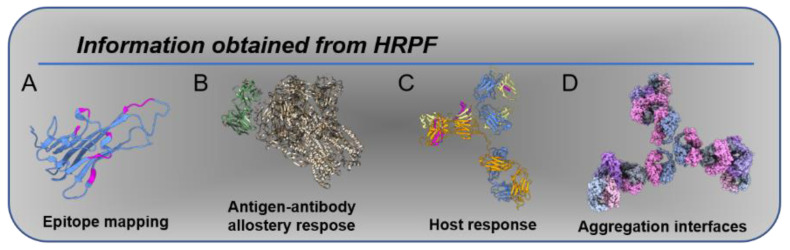
Information obtained from HRPF: (**A**) epitope mapping, as in [20]; (**B**) allosteric changes induced by antigen binding [24]; (**C**) host response, as in [23], in which anti-drug antibodies exhibit protection in response to therapy; (**D**) aggregation interfaces, as in [22].

**Figure 2 antibodies-11-00071-f002:**
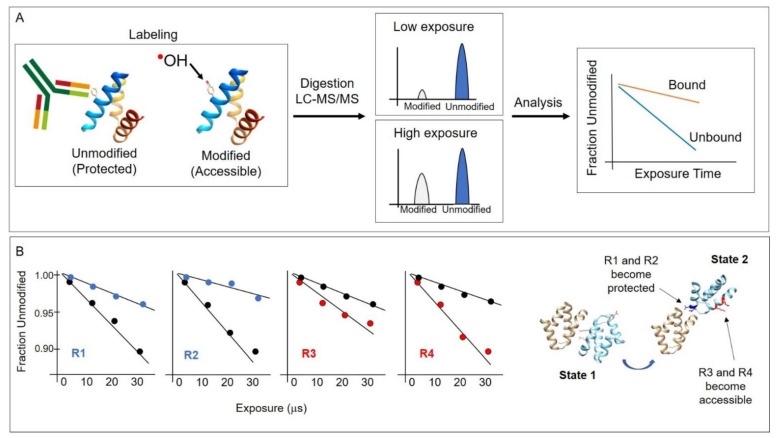
(**A**) Conceptual overview of the HRPF method. Hydroxyl radicals are generated through various means described in this article and covalently modify protein side chains that are not solvent protected, such as in an antibody–antigen interaction region. After hydroxyl radical labeling, proteins are digested, and LCMS is performed to identify sites of modification. Plotting the modification as a function of radical exposure reveals protected regions. Typically, the “fraction unmodified” is plotted per peptide or residue. (**B**) A hypothetical example to illustrate the application of the method. In this example, residues R1 and R2 become solvent protected, while residues R3 and R4 become solvent accessible when the protein conformation changes from one state to another, such as in the rotation of one domain relatively to another. The dose–response plots show the rate of modification differences between state 1 (black circles) and state 2 (blue or red circles) showing a decrease or increase in solvent accessibility, respectively.

**Figure 3 antibodies-11-00071-f003:**
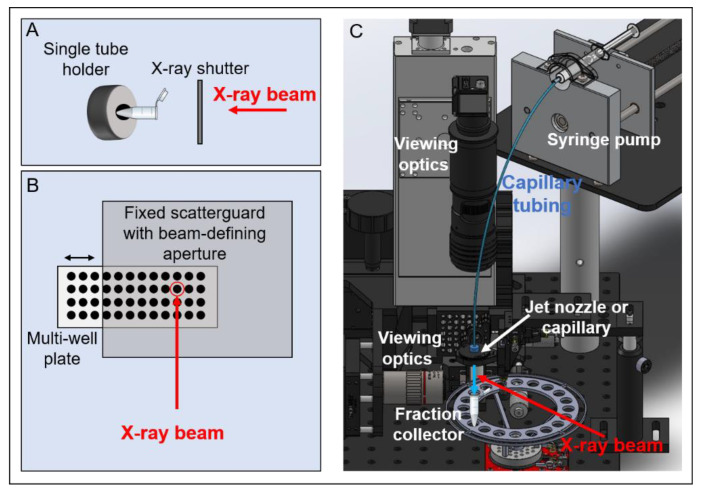
Overview of the synchrotron X-ray HRPF experiment sample delivery modes: (**A**) Single-tube exposure is defined by X-ray shutter opening/closing. (**B**) Well-plate velocity determines X-ray exposure times. For these methods of exposure, scavenging buffer is added post-exposure after tubes or plate is removed from the endstation. (**C**) Syringe pump delivers sample through a capillary or jet nozzle into a fraction collector containing scavenging buffer; velocity of sample and beam size determine exposure times. Other components of a typical XFMS experiment include pin diodes for flux monitoring, cooling blocks, optics for sample viewing, and beam-defining apertures and scatter guards.

**Figure 4 antibodies-11-00071-f004:**
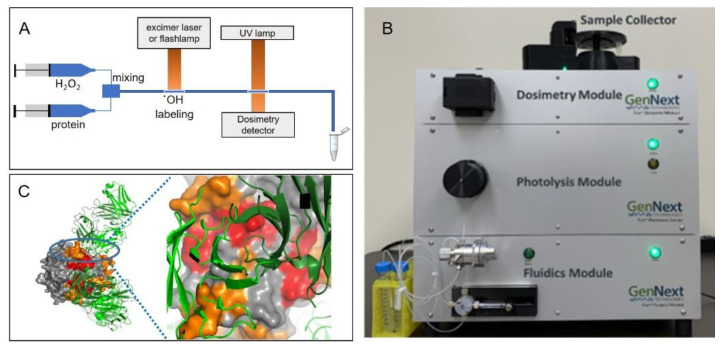
(**A**) Overview of the FPOP experiment. Protein is rapidly mixed with hydrogen peroxide; a UV laser or flashlamp is used to photolyze the hydrogen peroxide and produce hydroxyl radicals, which oxidatively modify accessible protein side chains. An optional inline dosimeter such as that described in [20] can be used to monitor dosage. Samples typically exit the apparatus into scavenging buffer. (**B**) A commercial FPOP system that uses a flashlamp for hydroxyl radical generation. (**C**) FOX system HRPF protection of TNFα plotted on the X-ray crystal structure of the TNFα-adalimumab Fab hexameric complex. Figure 4C is reproduced from [20], used with permission.

**Figure 5 antibodies-11-00071-f005:**
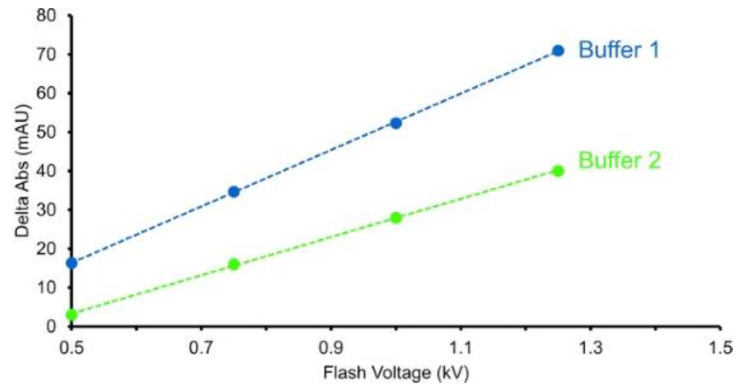
Adenine decreases 285 nm UV absorbance. The change in 265 nm absorbance is shown for different flash voltages and for two buffers. Buffer 2 contains a radical scavenger which decreases the change in adenine absorbance compared with buffer 1. Increased flash voltage is required to overcome the radical scavenging effects of buffer 2. Figure is adapted from [47], used with permission.

**Figure 6 antibodies-11-00071-f006:**
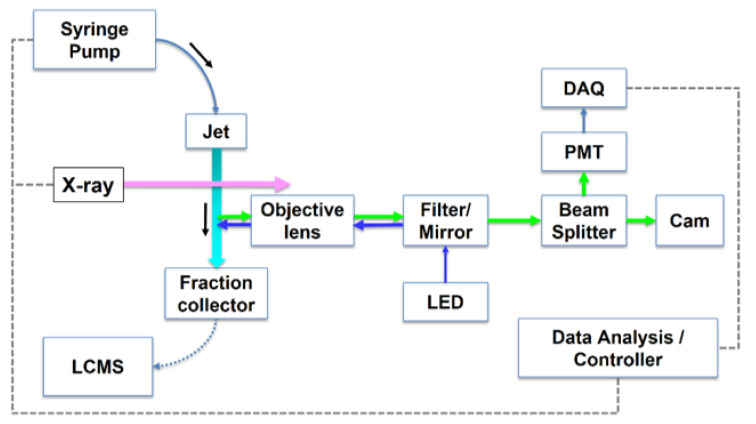
Conceptual diagram of inline dosimetry monitoring in XFMS experiments. Excitation light impinges on a flowing sample (capillary or jet delivery) immediately after X-ray exposure and before sample enters the fraction collector. The resulting fluorescence is collected along the same light path and analyzed to determine appropriate X-ray exposure values. Reproduced from [82], Figure 1A, used with permission.

**Table 1 antibodies-11-00071-t001:** Comparison between various OH-generation methods with buffer considerations, access, and timescales probed.

Method of OH Radical Generation	Sample/Buffer Considerations	Access	Timescale Accessible
Fenton reaction	Fe-EDTA and H_2_O_2_ added	Lab-based approach	Seconds to minutes
X-ray	No exogenous chemicals added	Synchrotron facilities (user proposals required)	10 microseconds to hours
UV	H_2_O_2_ added	Lab-based; UV laser required	1 microsecond to minutes
Plasma	H_2_O_2_ added	Lab-based (flash-lamp required) or through commercial venue	1 microsecond to minutes

## Data Availability

Not applicable.

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
