# Peer review of "Structural Investigation of Therapeutic Antibodies Using Hydroxyl Radical Protein Footprinting Methods"

_2073-4468, 2022, doi:10.3390/antib11040071_

Round 1

Reviewer 1 Report

It is a very nice review, compact and focused, on the application of hydroxyl radical protein footprinting (HRPF) for epitope mapping.  It is mostly devoted to the methods used to produce oxidative modifications.  Although comparison of HRPF to other methods of footprinting seems to be beyond the scope of the review, it would be very informative at least to mention the quality of the data obtained with HRPF.

Minor points:

- Abbreviation “LCMS” should be explained at the first appearance.

- Lines 359, 384: since adenine is a nucleobase, not a dosimeter, the phrase should be corrected.

Author Response

Overall:

We have added an overall description of the HRPF workflow, including how data is collected, and discussed its application relative to other standard structural biology methods on page 4, lines 146-175.

Minor points:

- Abbreviation “LCMS” should be explained at the first appearance.

The acronym has been spelled out at first appearance on page 3, line 127.

- Lines 359, 384: since adenine is a nucleobase, not a dosimeter, the phrase should be corrected.

We have specified that adenine is a nucleobase that can be used as a dosimeter on page 10, line 368.

Reviewer 2 Report

11)   Title of the review is suggesting antibody characterization but review only focused on hydroxy radical protein foot printing history and new advancements. Consider including HRPF application in structural characterization.

22)   Include a table comparing different methods of HRPF and its applications.

33)  Provide experiment Flow chart of HRPF protocol from protein to analysis. 

 4) For figure 1 references shows the application references. Discussing each reference would be helpful to showcase advantages of HRPF methods in solving complex problems. Only one  reference is related to antibody aggregation. Consider discussing and referring this publication https://www.ncbi.nlm.nih.gov/pmc/articles/PMC3564890/ (Structural analysis of a therapeutic monoclonal antibody dimer by hydroxyl radical footprinting)

55)  Include antibody basic structural information and post translational modifications. Antibody undergoes variety of post-translational modifications which impacts structure and function specially glycosylation. Extensively discussed in following articles discuss and include as references. https://www.ncbi.nlm.nih.gov/pmc/articles/PMC9021442/,

https://www.ncbi.nlm.nih.gov/pmc/articles/PMC6923169/

https://www.sciencedirect.com/science/article/pii/S135964461600026X#fig0010

66)  Can authors comment on limitations and opportunities of using hydroxy radical foot printing methods.

77)  Authors could include this publication as reference and discuss the application. Structural Analysis of the Glycosylated Intact HIV-1 gp120–b12 Antibody Complex Using Hydroxyl Radical Protein Footprinting | Biochemistry (acs.org)

88)  Does this method could apply to antibody drug conjugates.

Author Response

11) Title of the review is suggesting antibody characterization but review only focused on hydroxy radical protein foot printing history and new advancements. Consider including HRPF application in structural characterization.

The article gives 5 examples of the application of HRPF for antibody structural characterization in the introduction. In the revised version, we have now included 2 more examples in the introduction to showcase the application of HRPF to glycosylated proteins, and 3 more examples of using HRPF to mapping antibody interactions (page 2, lines 83-93). The article is not intended to cover in any depth previous antibody characterization studies, but rather to review the methods of HRPF and refer the reader to previous antibody studies of interest.

22) Include a table comparing different methods of HRPF and its applications.

A summary table has been added on page 6, indicating the method of OH generation, sample/buffer considerations, and timescales accessible by the methods. Applications have not been included specifically in the table, since the application areas of HRPF is similar for each of these methods of OH radical generation.

33)  Provide experiment Flow chart of HRPF protocol from protein to analysis. 

A concept figure has been added, and is now Figure 2 on page 5. A summary of the flow chart has been added on page 4, lines 146-158.

 4) For figure 1 references shows the application references. Discussing each reference would be helpful to showcase advantages of HRPF methods in solving complex problems. Only one  reference is related to antibody aggregation. Consider discussing and referring this publication https://www.ncbi.nlm.nih.gov/pmc/articles/PMC3564890/ (Structural analysis of a therapeutic monoclonal antibody dimer by hydroxyl radical footprinting)

This reference has been added, with a brief description of the work (page 2, lines 83-84). The references are given in the figure legend and are summarized in the introductory paragraph. The article is not intended to describe previous studies in any depth, but rather to briefly summarize previous studies so that readers may read the referenced articles if they wish to gain a more complete understanding of the application to specific protein systems.

55) Include antibody basic structural information and post translational modifications. Antibody undergoes variety of post-translational modifications which impacts structure and function specially glycosylation. Extensively discussed in following articles discuss and include as references. https://www.ncbi.nlm.nih.gov/pmc/articles/PMC9021442/,

https://www.ncbi.nlm.nih.gov/pmc/articles/PMC6923169/

https://www.sciencedirect.com/science/article/pii/S135964461600026X#fig0010

We've added the reference on glycoengineering to the introduction paragraph. This is an important aspect of therapeutic antibody development, though a review of antibody structure is not intended to be part of this review on HRPF methods.

66) Can authors comment on limitations and opportunities of using hydroxy radical foot printing methods.

A general discussion on practical implementation and limitations of HRPF has been added (page 4, lines 160-176).

77)  Authors could include this publication as reference and discuss the application. Structural Analysis of the Glycosylated Intact HIV-1 gp120–b12 Antibody Complex Using Hydroxyl Radical Protein Footprinting | Biochemistry (acs.org)

This reference has been added, with a brief description of the work (page 2, lines 85-86).

88) Does this method could apply to antibody drug conjugates.

This is an interesting point. Any characterization of antibody-antigen binding could help with design of antibody drug conjugates, and we have added a reference to a study in which this was done in the introduction. If the reviewer is referring to the characterizing the “footprint” of the drug, such as a cytokine, onto the antibody, then we are not aware of studies in this area. We have also added a reference to an article (Kiselar and Chance, 2018) which describes HRPF applied generally to protein drug interactions.

Reviewer 3 Report

The manuscript reviews recent advances in the field of hydroxyl radical protein footprinting with emphasis on practical considerations and advances. 

The reader is provided a good overview of applications areas and benefits of the method compared to other structural methods. However, in the main part of the review too many technical details are mentioned, e.g. row 207 -210 („the shutter material must withstand….“). Other parts sound like an advertising brochure (row 303-313).

 Following improvements are recommended:

- Reduction the technical details in the article and addition of a section concerning data analysis including identification and quantification of oxidation products

- addition of a statement to other practical considerations (e.g. time for method optimization, analysis and data interpretation)

- A more comprehensive review of published work relating dosimetry (e.g. reporter peptide versus small molecule)

- more meaningful figures:

               - Figure 1: is it possible to show how the MS raw data translate into the mapping of one example application? As no color code is given, it is difficult for the reader to follow the data interpretation from raw data to the color mapping.

               - Figure 3: 3C would improve if labels / color code meaning would be added. How good was the agreement between the epitope determined by FPOP compared to other methods?

               - Figure 4: 4A contains the raw data for „buffer 1“ in 4B. Therefore, it is not strictly necessary to show 4A. Better name the buffers 1 and 2 and the radical scavenger so that the reader can follow the reasoning. Alternatively, consider replacing the figure with a more conceptual diagramm showing what parameters impact hydroxyl radical availability and what possibilities are avilable to determine their impact.

- Typo: row 426 „fluorescence“

- References: use consistent page numbering (first & last page ?)

Author Response

- Reduction the technical details in the article and addition of a section concerning data analysis including identification and quantification of oxidation products

We have included technical details where needed to support or explain recent developments. For example, the X-ray shutter material and limitation is important to include in the article, since this technical detail is exactly what led to the development of capillary/jet flow methods, the subject of section 4.1.3. We have added a sentence to this effect on page 7, line 281. An in-depth description on data analysis is outside the scope of this mini-review, though we have added a conceptual figure (now Figure 2) and a general description of data analysis (page 4, lines 146-158).

- addition of a statement to other practical considerations (e.g. time for method optimization, analysis and data interpretation)

Practical considerations been added on page 4, lines 159-176.

- A more comprehensive review of published work relating dosimetry (e.g. reporter peptide versus small molecule)

A review of dosimetry work is given in Xie & Sharp 2015 and in McKenzie-Coe et al 2022. We have added a sentence referring the reader to these reviews on page 11, line 445-446.

- more meaningful figures:

               - Figure 1: is it possible to show how the MS raw data translate into the mapping of one example application? As no color code is given, it is difficult for the reader to follow the data interpretation from raw data to the color mapping.

These figures are summarized on page 2, lines 71-83. The figure is meant to provide a very general overview of the types of applications amenable to the HRPF method. We have also added a conceptual figure (Figure 2) showing how LCMS data can be mapped onto a protein structure.

               - Figure 3: 3C would improve if labels / color code meaning would be added. How good was the agreement between the epitope determined by FPOP compared to other methods?

This figure is reproduced from a previous publication, and we refer the reader to that publication for in-depth analysis of the data. The agreement between FOX/FPOP data and the previously published XRC data was excellent. All protected peptides were within the TNFalpha epitope, with only one epitope peptide unidentified in FOX/FPOP due to lack of labeling of the peptide. No false negatives or false positives were detected in the FOX/FPOP data in relation to the XRC data.

               - Figure 4: 4A contains the raw data for „buffer 1“ in 4B. Therefore, it is not strictly necessary to show 4A. Better name the buffers 1 and 2 and the radical scavenger so that the reader can follow the reasoning. Alternatively, consider replacing the figure with a more conceptual diagramm showing what parameters impact hydroxyl radical availability and what possibilities are avilable to determine their impact.

- Typo: row 426 „fluorescence“

This is now fixed

- References: use consistent page numbering (first & last page ?)

This now fixed

Reviewer 4 Report

Summary
Antibody Therapeutics is one of the fastest growing therapeutic areas. HRPF is a effective mass-spec based technique for structural characterization of mAbs since it has quite a few unique benefits over standard structural methods. It is significantly faster and more cost effective than x-ray crystallography and it does not have a protein size limitation unlike NMR. This review paper highlights the application of HRPF for antibody structural characterization and discusses different methods of generating OH radicals to perform the footprinting experiments.

General concept comments

A figure showing HRPF workflow would be useful along with the discussion in lines 113-116.

A discussion about which amino acids show detectable modification by standard HRPF methods would be beneficial for the readers to determine if the resolution of the HRPF dataset would be appropriate for their specific applications.

Line 221-228: A discussion about which amino acids are modified by CF3 vs OH and a comparison of CF3 labeled protein compared to a protein labeled with OH alone would strengthen this section.

The paper is missing examples of HRPF data (% change in modifications of peptides/residues). Figures showing HRPF data and a discussion of the data interpretation should be included.

A discussion of the limitations of HRPF, specifically the limited resolution due to low reactivities with certain amino acids is important

Author Response

A figure showing HRPF workflow would be useful along with the discussion in lines 113-116.

A concept figure has been added, and is now Figure 2 on page 5. We have added a description of the HRPF workflow and discussed its application relative to other standard structural biology methods on page 4, lines 146-176.

A discussion about which amino acids show detectable modification by standard HRPF methods would be beneficial for the readers to determine if the resolution of the HRPF dataset would be appropriate for their specific applications.

Page 4, lines 187-189 lists the most commonly modified residues. We have added a sentence on the relative reactivity of amino acids to hydroxyl radical, page 4 lines 189-193.

Line 221-228: A discussion about which amino acids are modified by CF3 vs OH and a comparison of CF3 labeled protein compared to a protein labeled with OH alone would strengthen this section.

We have added a sentence describing reactivities, and made explicit reference to a recent article on CF3 labeling, so that the reader can learn more details about CF3 vs OH labeling (page 8, lines 304-306).

The paper is missing examples of HRPF data (% change in modifications of peptides/residues). Figures showing HRPF data and a discussion of the data interpretation should be included.

A concept figure has been added, and is now Figure 2 on page 5. This figure includes a hypothetical HRPF study, showcasing how dose response plots appear, general level of modifications, and how this data is mapped onto a structure.

A discussion of the limitations of HRPF, specifically the limited resolution due to low reactivities with certain amino acids is important

We have added sentences describing reactivities: page 4 lines 189-193 and page 8 lines 304-306.

Round 2

Reviewer 2 Report

Thank you for addressing the request changes .